# Influence of Heat–Cool Cyclic Exposure on the Performance of Fiber-Reinforced High-Strength Concrete

Ibrahim Hakeem [1,*], Md. Akter Hosen [2], Mana Alyami [1], Shaker Qaidi [3,4,*] and Yasin Özkılıc [5]

1 Department of Civil Engineering, College of Engineering, Najran University, Najran 55461, Saudi Arabia
2 Department of Civil and Environmental Engineering, College of Engineering, Dhofar University, Salalah 211, Oman
3 Department of Civil Engineering, College of Engineering, University of Duhok, Duhok 42001, Iraq
4 Department of Civil Engineering, College of Engineering, Nawroz University, Duhok 42001, Iraq
5 Department of Civil Engineering, Faculty of Engineering, Necmettin Erbakan University, Konya 42000, Turkey
* Correspondence: iyhakeem@nu.edu.sa (I.H.); shaker.abdal@uod.ac (S.Q.)

**Abstract:** Sometimes civil engineering infrastructures have been constructed in hot and cold weathering regions such as desert areas. In such situations, the concrete is not only smashed by hot and cold processes but also spoiled by shrinkage cracking. Therefore, this study intends to examine the influence of heat–cool cycles on high-strength concrete comprising various fibers, such as natural date palm, polypropylene, and steel fibers, and their different volume percentages. The most popular technique for improving the structural behavior of concrete is fiber insertion. Fibers decrease cracking occurrences, enhance early strength under impact loads, and increase a structure's ability to absorb additional energy. The main goal is to examine the effects of three different types of fibers on regular concrete exposed to heat–cool cycles. For each type of fiber, three dosages of 0.2%, 0.6%, and 1% were used to create high-strength concrete. After 28 days of regular water curing and six months of exposure to heat-and-cold cycles, all specimens were tested. The heat–cool cycles entailed heating for two days at 60 °C in the oven and cooling for another two days at room temperature. The results of the experiment showed that fiber reinforcement in concrete improves its strength and durability. The flexural strength was substantially improved by increasing the date palm, polypropylene, and steel fibers into the high-strength concrete with and without heat–cool cycles. Adding increments of date palm, polypropylene, and steel fibers into high-strength concrete revealed a significant improvement in energy absorption capacity in both cases, i.e., with or without the implementation of heat–cool cycles. Therefore, the natural date palm fibers might be utilized to produce sustainable fibrous high-strength concrete and be applicable in severe weathering conditions.

**Keywords:** fibrous concrete; date palm fiber; engineering characteristics; durability; thermal cycles; energy absorption capacity

## 1. Introduction

Concrete structures are typically exposed to a variety of diverse environmental conditions throughout their lifespan [1–3]. Hence, a concrete structure's resilience is measured by how well it can resist certain exposure conditions without needing extensive maintenance or rehabilitation [4,5]. Concrete is said to be a composite material that can sustain itself for many years, or even centuries, with little to no maintenance [6]. Without reinforcing, plain cement concrete is made up of cement, fine, and coarse aggregate. To produce different types of concrete that are appropriate for diverse structural loading and environmental conditions, changes can be made to the ingredients used to generate plain cement concrete (PCC) and its mix design. Several performance-related difficulties are presented in order to emphasize the typical concrete's poor performance even further [7,8].

"The advent of ultra-high-performance concrete (UHPC) in the 1990s was a technological improvement in the creation of concrete [9,10]. High compressive strength, high tensile

strength, and a good degree of fracture toughness and ductility were the distinguishing characteristics of this innovative concrete [11–13]. Additionally, fibers are being employed as a discrete three-dimensional reinforcement to overcome PCC's shortcomings and as a substitute for UHPC [14–16]. In order to increase its resistance to loads, fiber-reinforced concrete (FRC) inserts fiber into its composition. Different kinds of FRC have been developed, each with unique and significant benefits. FRCs have a wide range of applications because of their many benefits—good tensile strength, ductility, and fatigue resistance—which include building pavements, industrial floors, tunnel linings, slope stabilization, and impact-resistant constructions, among others" [17,18].

The initiation and spread of cracks in concrete under tensile and compressive loads can be prevented or delayed by using the right fiber type. Commercially accessible reinforcements come in a variety of categories and have features that make them suitable for particular applications. Examples include "carbon fiber [19,20], steel fiber (SF) [21,22], glass fiber [23], polypropylene fiber [24,25], organic fibers [26,27], carbon nanotubes [28], basalt fiber [29] and more. In comparison to other industrial fibers, SF is by far the better fiber when it comes to the mechanical performance of concrete. SF has a high elastic modulus of roughly 200 GPa, and a high tensile strength of over 1200 MPa. The literature has established a paradigm that supports the viability of SF as an excellent reinforcement material that ensures satisfactory tensile, compressive, flexural, and shear strength qualities [30,31]. Azad, A. K. et al. [32] expressed the experimental outcomes of the flexural test, the specimens can support more loads once they reach the cracking load; however, once they reach the peak load, a softening mode of collapse is seen, exhibiting great ductility. Additionally, it has been demonstrated that the use of steel fibers increases the reinforced concrete beams' resistance to shear failure, negating the need for stirrups" [33–35].

Mishra, S. et al. [36] have utilized numerous chemically altered sisal fibers as reinforcement, in addition to glass fibers, in the polyester matrix to improve the mechanical properties of the hybrid composites. According to the experimental findings, hybrid composites absorb less water than unhybridized composites. Mohanty, A. K et al. [37] demonstrated that these concrete specimens' water absorption rates were low when compared to un-palmed and categorized composites. When compared to well-known composites, such as glass and palm, bamboo and palm, and glass manufactured using the same techniques, an analysis of the tensile, flexural, and dielectric properties of composites revealed comparable results for characteristics, such as tensile strength. Priya, S. P. et al. [38] examined the tensile strength of these palm and glass composites and determined that adding more fabric to these composites improved their mechanical qualities. It was discovered that the matrix and the reinforcement had strong interfacial bonding and chemical resistance. Althoey, F. et al. [39] investigated the engineering characteristics of date palm fiber-infused high-strength concrete as well as the performance of conventional steel and polypropylene fibers. The concrete samples were fabricated using 0.0%, 0.20%, 0.60%, and 1.0% volumes of date palm, steel, and polypropylene fibers. The results revealed that 1% of date palm, steel, and polypropylene fibers boosted the splitting tensile strength by 17%, 43%, and 16%, respectively. For date palm, steel, and polypropylene fibers, the flexural strength was enhanced from 60% to 85%, 67% to 165%, and 61% to 79% respectively, by adding 1% fiber in comparison to the reference sample.

The main aim of this paper is to investigate the influence of three different types of fibers (date palm, polypropylene, and steel) in creating high-strength concrete under the heat–cool exposure cycles.

## 2. Research Significance

One of the most promising and cost-effective solutions is to replace traditional steel-reinforced concrete with fiber-reinforced concrete for structural applications. This study focuses on high-strength concrete comprising date palm fibers, which are agro-waste in the Arabian gulf region, and its properties compared to conventional polypropylene and steel fibers. Durability is the key concern for new date palm fiber-reinforced high-strength

concrete. Concrete's resilience and service life are largely influenced by environmental variables including heating–cooling and freezing–thawing cycles. One of the most prevalent and harmful variables for concrete in service is the heat–cool cycle, which is found in many parts of the world, including the Arabian Gulf, Northwest China, and southern California in the United States. Therefore, this study focuses on the influence of heat–cool cyclic exposure on the performance of high-strength concrete comprising date palm fibers and conventional polypropylene and steel fibers.

### 3. Materials and Methods

*3.1. Materials*

3.1.1. Cement

In this investigation, the high-strength date palm, polypropylene, and steel fiber concrete specimens were made using regular Portland cement Type-I. The cement had a fineness of 4100 cm$^2$/g and a specific gravity of 3.15. According to the manufacturer [40] and verified by ASTM C 150 [41], the cement included 59% C3S, 12.10% C2S, 10.60% C3A, and 10.4% C4AF. Table 1 includes information about the cement's chemical arrangement.

**Table 1.** Chemical arrangement of cement.

| Chemical Composite | Cao | Al$_2$O$_3$ | Fe$_2$O$_3$ | MgO | SiO$_2$ | SO$_3$ | LOI | K$_2$O | Insoluble |
|---|---|---|---|---|---|---|---|---|---|
| Mass (%) | 63.83 | 6.25 | 3.45 | 0.97 | 19.70 | 2.25 | 1.52 | 1.08 | 0.95 |

3.1.2. Aggregates

Natural dune sand was utilized as a fine aggregate, with the majority of its particles passing through a 4.75 mm sieve [42]. The high-strength date palm, polypropylene, and steel fibrous concrete was made utilizing crushed stone with a maximum size of 20 mm as the coarse aggregate. Table 2 demonstrates the physical characteristics of the fine and coarse aggregates.

**Table 2.** Physical characteristics of the aggregates.

| Characteristics | Type of Aggregate | |
|---|---|---|
| | Fine | Coarse |
| Bulk Density (kg/m$^3$) | 1535.74 | 1630.00 |
| Specific Gravity | 2.67 | 2.77 |
| Fineness Modulus | 2.23 | 7.34 |
| Water Absorption (%) | 1.31 | 0.69 |

3.1.3. Superplasticizer and Water

For the production of high-strength date palm, polypropylene, and steel fibrous concrete, super plasticizers (SP) are renowned as excellent water reducers. In this study, Glenium® 110M, which is based on polycarboxylate ether, was applied as an SP when fabricating the fibrous concrete.

In both cases, filtered tap water was a key component in the production and curing of high-strength fibrous concrete. The requirements for producing high-strength fibrous concrete with ASTM C1602/C1602M [43] have been related to the qualities of water.

3.1.4. Date Palm Fibers

The date palm fibers were sourced from date palm trees in and around Najran, Saudi Arabia, that were 15 to 25 years old. These trees represent one of the best accessible diversities and are responsible for a sizable amount of agricultural waste products. Using

a manual process, the date palm fibers were manually removed and collected from the palm trees. The bidirectional date palm fibers are positioned around the trunk of the tree and consist of two or three layers that are packed and overlaid. The different lengths and diameters of the raw date palm fibers were collected from the agro-farm, as shown in Figure 1.

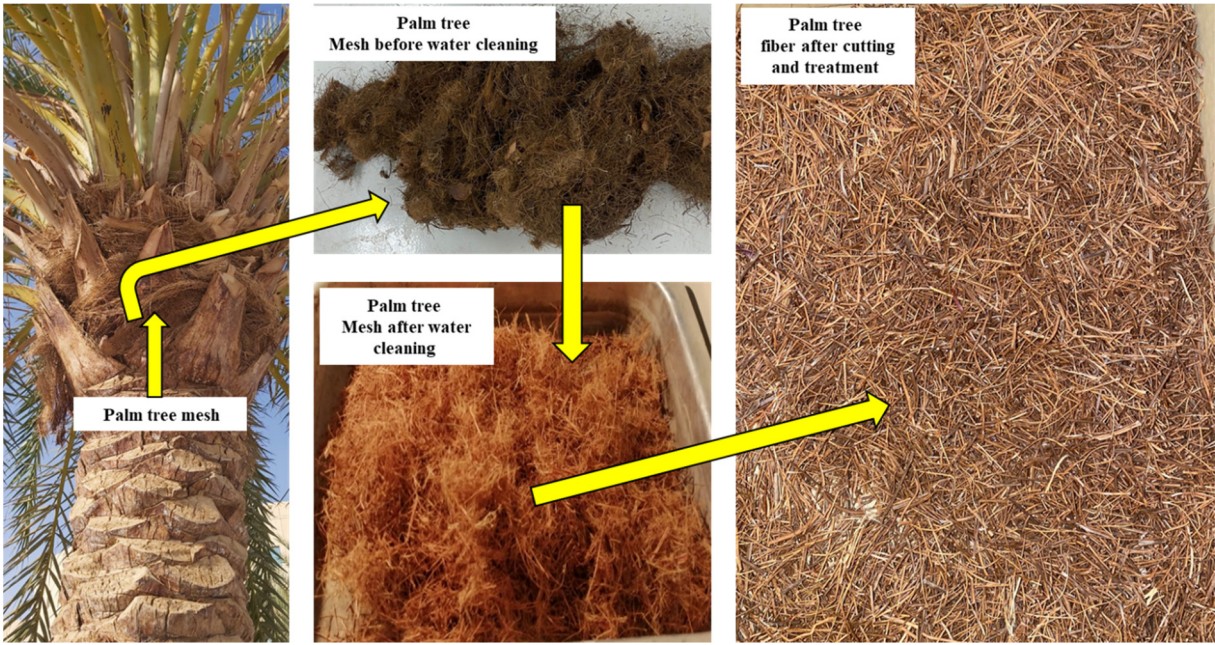

**Figure 1.** Collecting procedure of date palm fibers as a waste material.

Chemical Treatment of Date Palm Fibers

"To remove any potential contaminants from the surface of the fibers and to improve the compatibility of the fibers with other concrete ingredients, the date palm fibers were chemically cured using varying concentrations of Sodium Hydroxide (NaOH). The major modification caused by this treatment is the disruption of hydrogen bonding within the network structure, here with increasing surface roughness. Here, aqueous NaOH is used to remove the lignin, wax, and oils from the cell walls. Thus, alkaline treatment often affects the cellulosic fibril, the degree of polymerization, and therefore the extraction of lignin and other non-cellulosic compounds. The treatment of date palm fibers was performed by NaOH solution immersion. The fibers were treated by immersing them individually in 1.5%, 3.0%, and 6.0% of NaOH solution. The fibers were immersed in the solution for 24 h at room temperature. Based on the effect on the fibers, a treatment with 3% NaOH was chosen due to the highest tensile strength of the fibers". Table 3 catalogs the physical characteristics of date palm fibers.

**Table 3.** Date palm fiber's physical characteristics.

| Type of Date Palm Fibers | Diameter, (mm) | Length (mm) | Elongation (%) | Strain | Tensile Strength (MPa) |
|---|---|---|---|---|---|
| Raw Fibers | 0.90 | 92 | 4 | 0.044 | 100 |
| 1.5% NaOH Treated | 0.65 | 79 | 6 | 0.058 | 174 |
| 3.0% NaOH Treated | 0.61 | 79 | 6 | 0.062 | 234 |
| 6.0% NaOH Treated | 0.69 | 80 | 5 | 0.055 | 181 |

### 3.1.5. Polypropylene Fibers

Polypropylene fiber was applied to manufacture the high-strength fibrous concrete compared with date palm and steel fibrous concrete. The manufacturer provided the polypropylene fiber's physical characteristics, which are presented in Table 4. Figure 2 shows the polypropylene fibers that were utilized in this investigation.

**Table 4.** Polypropylene fiber's physical characteristics.

| Length (cm) | Diameter (cm) | Density (g/cm³) | Young Modulus (GPa) | Elongation at Breaking (%) | Tensile Strength (MPa) |
|---|---|---|---|---|---|
| 1.20 | 0.0025 | 0.91 | 5.4 | 30 | 550 |

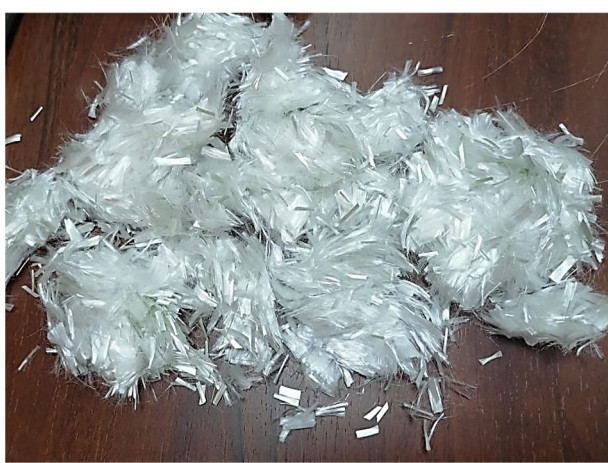

**Figure 2.** Polypropylene fibers.

### 3.1.6. Steel Fibers

The steel fibers were bundled with adhesive and hooked at both ends, as shown in Figure 3. The high-strength fibrous concrete was produced using these bundled steel fibers. Table 5 lists the physical characteristics of the steel fibers. Table 6 reveals the range of fibers used to manufacture the fibrous concrete, which ranges from 0% to 1.0% of the volume of the concrete.

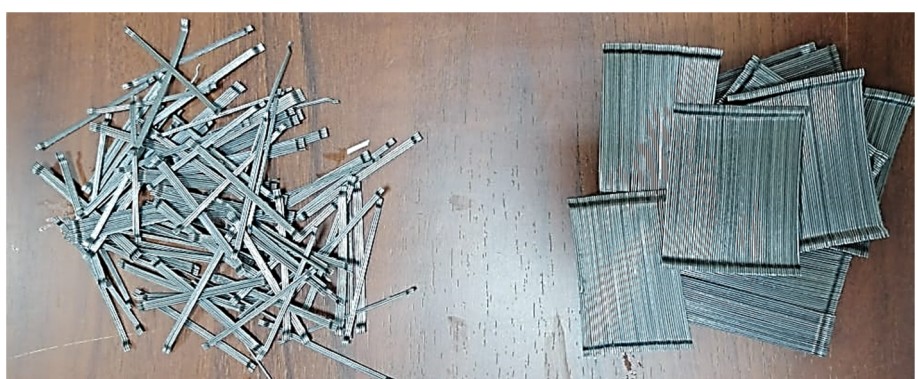

**Figure 3.** Bundled steel fibers.

**Table 5.** Steel fiber's physical characteristics.

| Length (cm) | Diameter (cm) | Aspect Ratio | Density (g/cm³) | Tensile Strength (MPa) |
|---|---|---|---|---|
| 6.0 | 0.075 | 80 | 7.85 | 625 |

**Table 6.** Fibrous concrete mix proportion.

| Mix ID | (%) | Fibers | | | Aggregates | | Cement | Water | SP |
|---|---|---|---|---|---|---|---|---|---|
| | | Date Palm | PP | Steel | Fine | Coarse | | | |
| | | Kilogram per Cubic Metre | | | | | | | |
| REF | 0 | - | - | - | | | | | |
| DF0.2 | 0.2 | 8 | - | - | | | | | |
| DF0.6 | 0.6 | 24 | - | - | | | | | |
| DF1.0 | 1.0 | 40 | - | - | | | | | |
| PF0.2 | 0.2 | - | 8 | - | 736.93 | 1105.40 | 400.00 | 176.40 | 2.0 |
| PF0.6 | 0.6 | - | 24 | - | | | | | |
| PF1.0 | 1.0 | - | 40 | - | | | | | |
| SF0.2 | 0.2 | - | - | 8 | | | | | |
| SF0.6 | 0.6 | - | - | 24 | | | | | |
| SF1.0 | 1.0 | - | - | 40 | | | | | |

*3.2. Methodology*

3.2.1. Mix Design and Specimens Preparation

Various tests that focus on specific aspects of the date palm, polypropylene, and steel fibrous high-strength concrete were used to evaluate the performance of the fibers. Different fibers (date palm, polypropylene, and steel) are blended with concrete in varying amounts. In this study, 0.2%, 0.6%, and 1.0% of fibers by volume of concrete have been applied for manufacturing the high-strength fibrous concrete. A total of ten mixtures were developed using date palm fiber (DF), polypropylene fiber (PF), and steel fiber (SF) as shown in Table 6.

The experimental program was conducted to calculate the various hardened features of the high-strength fibrous concrete using 100 mm cubes, 150 mm diameter × 300 mm height cylinders, and 100 mm × 100 mm × 500 mm prisms. Each mixture of high-strength fibrous concrete contained three specimens, and average values are presented in this study.

3.2.2. Thermal Cycles Procedure

The specimens were exposed to heat–cool cycles after 28 days of water curing by placing them into an oven. A single heating and cooling process cycle consisted of two days of heating at 60 °C and then two days of cooling specimens at room temperature 25 ± 5 °C, repeated for 180 days. This heat–cool cycle was chosen to simulate several areas of the Kingdom of Saudi Arabia's daily variation of ambient temperature, which is prevalent in the summer. Sufficient space between specimens was maintained to allow a uniform flow of hot air during heating and easy dissipation of heat during cooling. The specimens were placed at an adequate distance from the source of hot air so that the heat would not concentrate only on the surface of the specimens, as shown in Figure 4. After 180 days of exposure, the hardened density, compressive strength, flexural strength, ultrasonic pulse velocity (UPV), and water absorption of the specimens were determined.

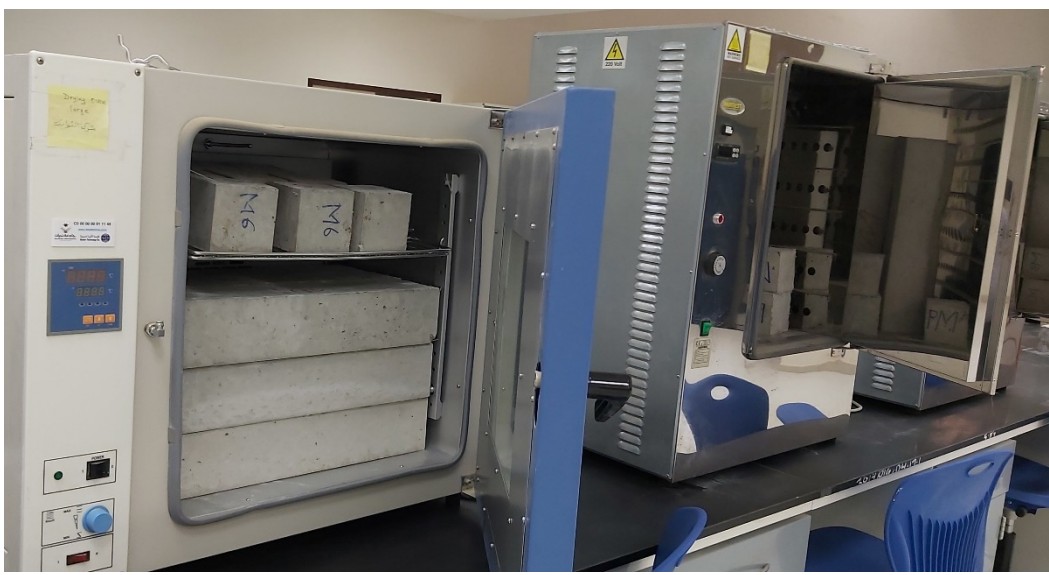

**Figure 4.** Fiber reinforced concrete specimens under heat–cool cycles.

### 3.2.3. Investigation of Structural Properties

Compressive Strength Test

The capacity of concrete to carry compressive loads till failure is known as compressive strength. Based on ASTM C109 [44], the compressive strength test for high-strength date palm, polypropylene, and steel fiber reinforced concrete was carried out. The $100 \times 100 \times 100$ mm cube specimens were evaluated after being hydrated for 28 days with ordinary drinkable water. The following formula was used to determine the compressive strength of the high-strength fibrous concrete specimens:

$$f'_c = \frac{F}{A_c} \tag{1}$$

where $f'_c$ is compressive strength in MPa; $F$ is the maximum compressive load at failure in N; $A_c$ is the cross-sectional area of the specimens in mm$^2$.

Flexural Strength Test

According to ASTM C293 [45], a center-point loading arrangement was employed to test the flexural strength of high-strength date palm, polypropylene, and steel fiber concrete prism specimens. This test was performed using a Universal Instron machine with a 400 kN loading capacity and a constant loading rate of 0.0167 mm/s. This experiment evaluated the high-strength date palm, polypropylene, and steel fibrous concrete's modulus of rupture (MOR). This method is frequently used to succeed in high-strength fibrous concrete's flexural strength. A linear variable displacement transducer (LVDT) installed at the center of 100 mm × 100 mm × 500 mm high-strength fiber reinforced concrete prism specimens were used to assess the displacement. While conducting the experiment on the prism specimens, the applied load and displacement were automatically recorded in the data logger. To study the load-displacement curves, the data logger's recorded results were transferred to a computer.

Density Test

After a 28-day curing period, the density of the high-strength date palm, polypropylene, and steel fiber concrete specimens was measured with reference to ASTM C138 [46]. Before performing the mechanical compression test, this experiment was performed on the specimens. The weight and volume of the fiber concrete specimens were assessed throughout these studies.

Water Absorption Capacity Test

The presence of enclosing little holes that are changed by excessive water is a sign of concrete's greater quality. As a result, "concrete quality measurements like density, stiffness, and durability are typically computed using the experiment to determine water absorption capacity. After meeting the requirement of the curing period of 28 days, the water absorption test for high-strength date palm, polypropylene, and steel fiber reinforced concrete was carried out in accordance with BS 2011 Part 122 [47] using cylindrical specimens with sizes of 75 mm in diameter and 150 mm in height. The high-strength fiber reinforced concrete specimens were initially dried for the first 72 h in an electric power oven at a constant temperature of 105 °C. The specimens were then removed from the oven, allowed to cool for 24 h in a dry environment, and weighed. The specimens were placed right away in a water tank at a temperature of 20 °C. The specimens were submerged in water for 30 min with the specimen's longitudinal axis kept horizontal. Following the collection of the specimens from the water, the clothing was dried to obtain a saturated surface state before being reweighed". The increase in weight caused by immersion in water, which was shown as a percentage of the dry weight of the specimen, was used to determine the water absorption capacity of fiber reinforced concrete specimens.

Ultrasonic Pulse Velocity (UPV) Test

The ultrasonic pulse velocity (UPV) test was used to validate the integrity and homogeneity of the high-strength date palm, polypropylene, and steel fiber reinforced concrete specimens [48]. The test was performed using the high-strength fiber reinforced concrete specimens in accordance with ASTM C597 [49].

Energy Absorption Capacity

The energy retained by the unit cross-sectional area at any displacement terminal point is used to represent the high-strength date palm, polypropylene, and steel fiber-reinforced concrete specimens' ability to absorb energy [50]. By using the area under the load vs. deflection graphs up to the specimens' rupture, the energy absorption capacity of the specimens was calculated.

## 4. Results and Discussions

### 4.1. Compressive Strength

The experimental outcomes of the compressive strength for date palm, polypropylene, and steel fiber-reinforced concrete specimens with and without implementing heat–cool cycles are shown in Figure 5. The compressive strength gradually enhanced with increasing the fiber content for date palm, polypropylene, and steel fiber-reinforced concrete specimens without applying the heat–cool cycles compared with the reference specimen, as shown in Figure 5a. At the same time, the compressive strength for specimens that underwent the heat–cool cycles was slightly reduced as the volume of fibers increased for date palm and polypropylene fiber-reinforced concrete specimens, but not steel fiber specimens, compared with reference specimens. Therefore, the compressive strength might predominantly rely on the strength of the aggregates [51], whereas the impact of heat–cool cycles on the strength may be trivial.

Figure 5b–d show "the correlation between the compressive strength and date palm, polypropylene, and steel fibers contents, respectively for the application of heat-cool cycles and without heat-cool cycles. This correlation demonstrated strong $R^2$ values for date palm, polypropylene, and steel fiber-reinforced high-strength concrete specimens". The predicted equations for high-strength date palm, polypropylene, and steel fiber-reinforced concrete specimens for the implementation of heat–cool cycles and without heat–cool cycles are specified by:

$$f'_{cDF} = 0.72v_f + 69.84 \; for \; date \; palm \; fibers \; with \; heat - cool \; cycles \qquad (2)$$

$$f'_{cDF} = 5.99v_f + 63.01 \; for \; date \; palm \; fibers \; without \; heat - cool \; cycles \tag{3}$$

$$f'_{cPF} = 5.13v_f + 67.13 \; for \; polypropylene \; fibers \; with \; heat - cool \; cycles \tag{4}$$

$$f'_{cPF} = 3.32v_f + 65.54 \; for \; polypropylene \; fibers \; without \; heat - cool \; cycles \tag{5}$$

$$f'_{cSF} = 0.44v_f + 73.69 \; for \; steel \; fibers \; with \; heat - cool \; cycles \tag{6}$$

$$f'_{cSF} = 7.57v_f + 62.51 \; for \; steel \; fibers \; without \; heat - cool \; cycles \tag{7}$$

where $f'_{cDF}$, $f'_{cPF}$, and $f'_{cSF}$ are the compressive strength (MPa) of date palm, polypropylene, and steel fiber-reinforced concrete specimens, respectively and $v_f$ is the fibers content (%) in the concrete specimens.

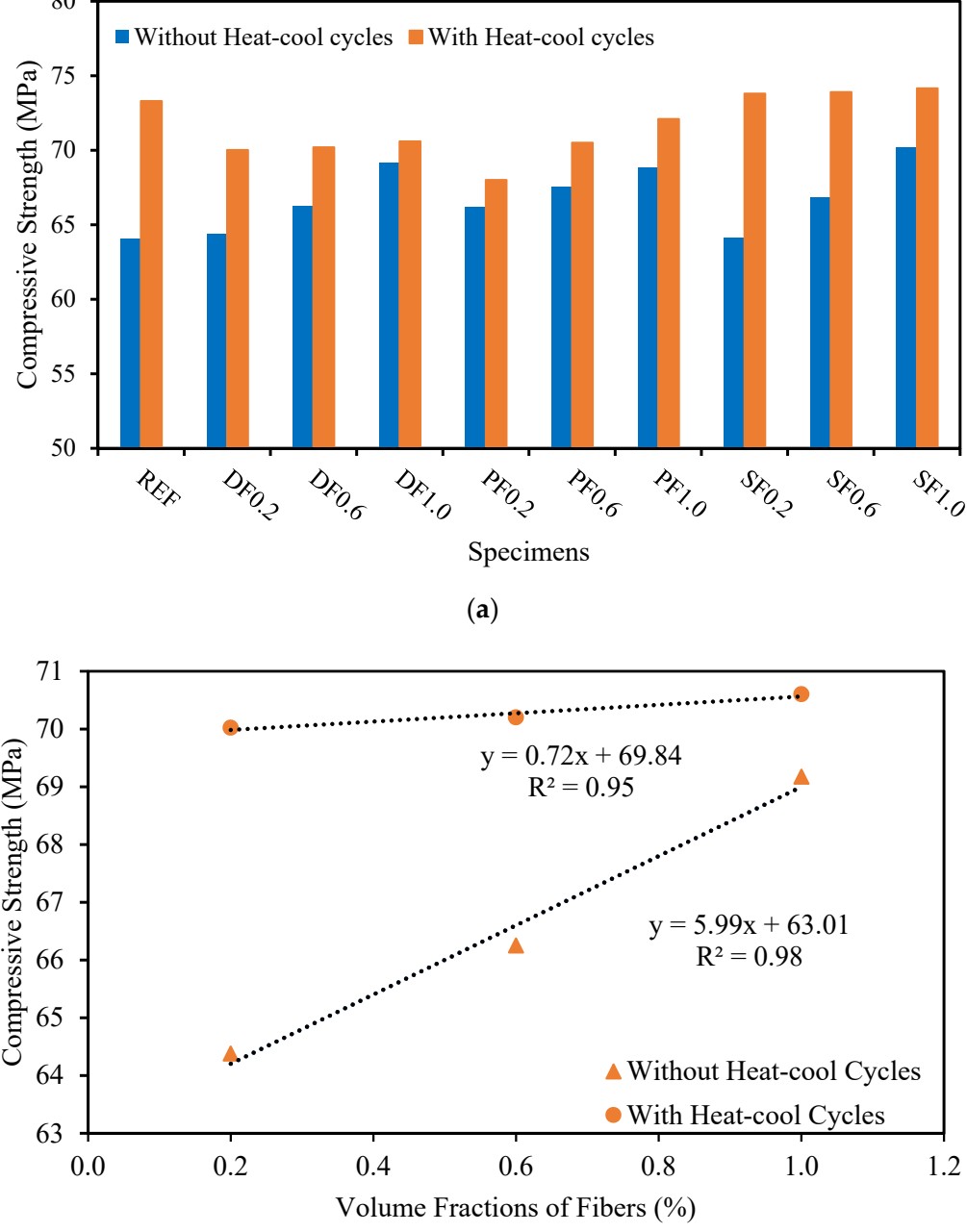

**Figure 5.** *Cont.*

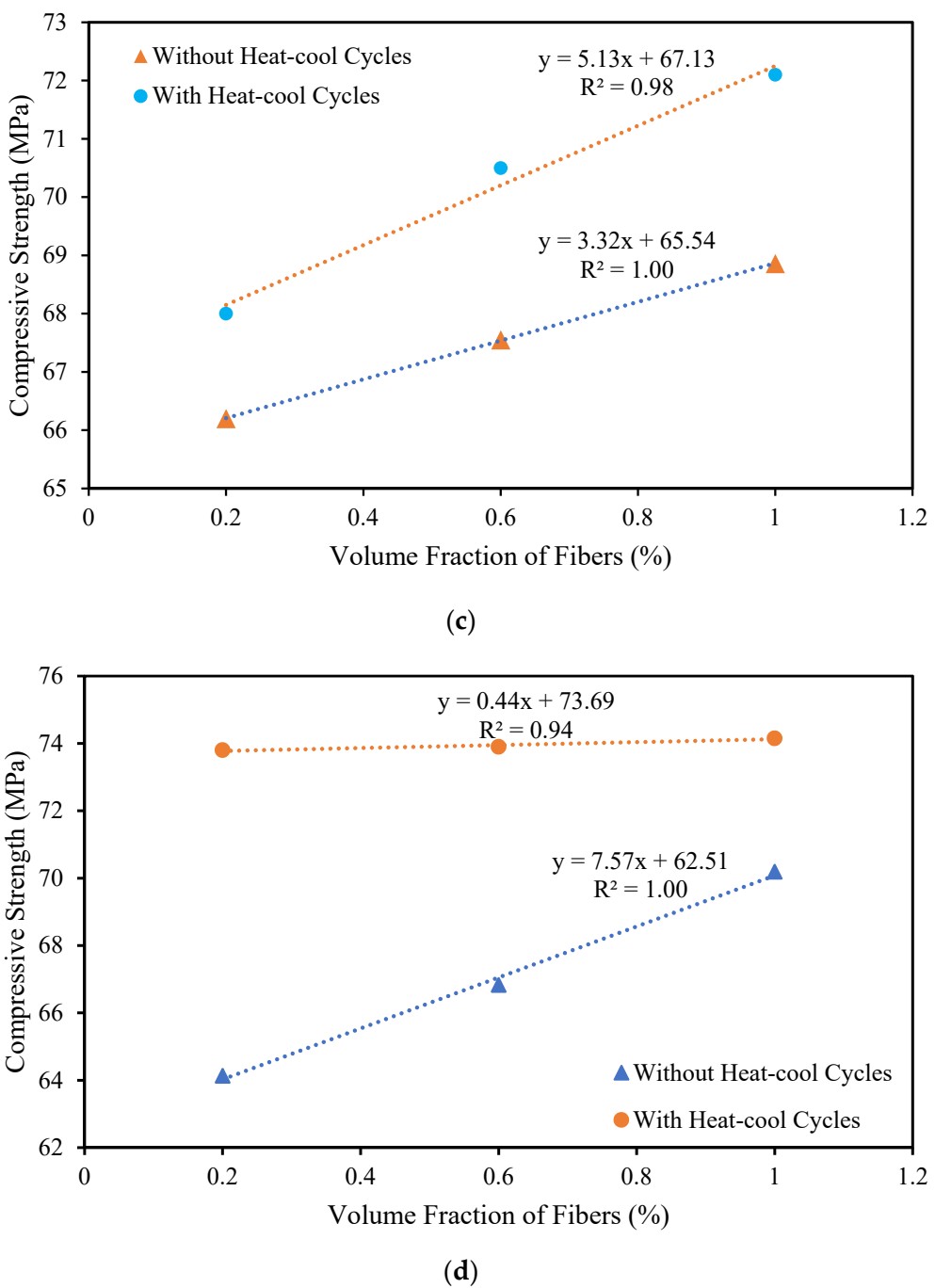

**Figure 5.** Influence of heat–cooling cycles on the compressive strength of fibrous concrete. (**a**) Assessment of compressive strength for heat–cool cycles; (**b**) Correlation between the compressive strength and amount of date palm fibers; (**c**) Correlation between the compressive strength and amount of polypropylene fibers; (**d**) Correlation between the compressive strength and amount of steel fibers.

### 4.2. Flexural Strength

The flexural performance of fiber-reinforced concrete is crucial for safeguarding infrastructures against severe weathering actions, such as freeze–thaws and extreme temperature [52–54]. The flexural strength of high-strength concrete comprising date palm, polypropylene, and steel fibers and its improvement are presented in Figure 6a. The addition of date palm, polypropylene, and steel fibers from 0% to 1% into the high-strength concrete significantly enhanced the flexural strength up to 85%, 79%, and 165%, respectively, compared with the reference specimen, without the implementation of heat–cool

cycles. This was increased to 4%, 2%, and 34% for date palm, polypropylene, and steel fibers containing specimens, respectively, compared with the reference specimen when heat–cool cycles were applied on the specimens. In both cases, the steel fibers significantly enhanced the flexural strength compared with the date palm and polypropylene. The steel fibers might withstand or postpone the arising initial cracks in the cross-section of the specimens because of their higher flexural rigidity and higher capability to resist the severe weathering action. On the other hand, date palm fibers exhibited better flexural performance over polypropylene fibers because of the larger length of fibers.

Whereas, Kriker, A. et al. [55] investigated the mechanical and physical characteristics of four different varieties of date palm fibers. The attributes of date palm fiber-reinforced concrete are also provided as a function of curing in water and in a hot, dry climate, including strength, continuity index, toughness, and microstructure. When hot, in both dry and water curing, it was discovered that increasing the length and percentage of fiber reinforcement improved the post-crack flexural strength and toughness coefficients but decreased the first crack and compressive strengths.

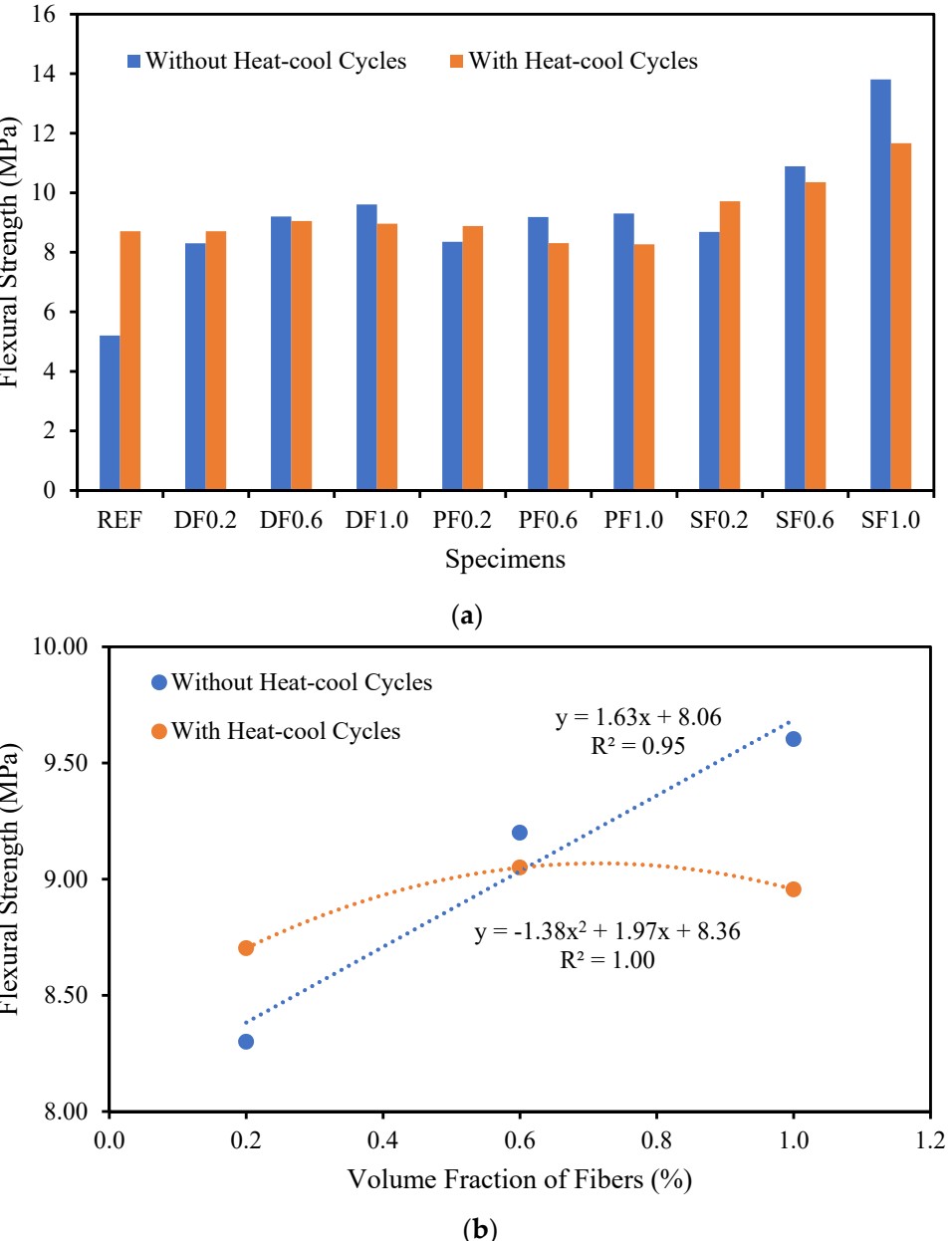

**Figure 6.** *Cont.*

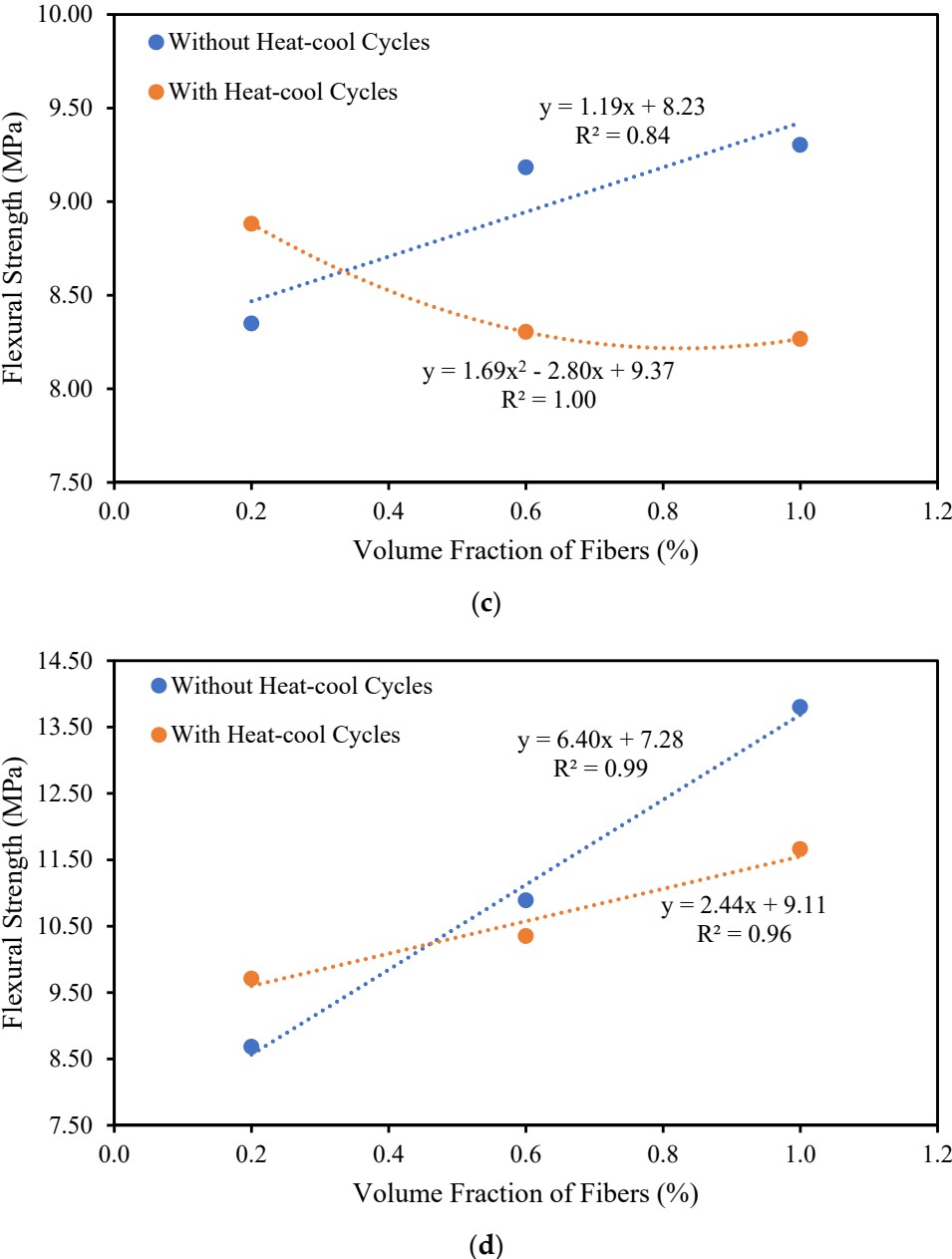

(**c**)

(**d**)

**Figure 6.** Impact of heat–cool cycles on the flexural strength of fibrous concrete. (**a**) Flexural performance of fibrous concrete changed due to heat–cool cycles; (**b**) Correlation between the flexural strength and amount of date palm fibers; (**c**) Correlation between the flexural strength and amount of polypropylene fibers; (**d**) Correlation between the flexural strength and amount of steel fibers.

"The relationship between the flexural strength and volume fraction of date palm, polypropylene, and steel fibers with applying the heat-cool cycles and without heat-cool cycles are displayed in Figure 6b–d, respectively. This correlation demonstrated linearly enhancing flexural strength for date palm, polypropylene, and steel fiber-reinforced high-strength concrete specimens without applying heat-cool cycles. By contrast, only steel fibers comprising specimens exhibited predominantly improving flexural strength under the heat-cool cycles". The flexural strength predicted equations for high-strength date palm, polypropylene, and steel-fiber-reinforced concrete specimens for the implementation of heat–cool cycles and without heat–cool cycles are specified by:

$$f_{rDF} = -0.32v_f^2 + 1.97v_f + 8.36 \ for \ date \ palm \ fibers \ with \ heat-cool \ cycles \qquad (8)$$

$$f_{rDF} = 1.63v_f + 8.06 \ for \ date \ palm \ fibers \ without \ heat-cool \ cycles \qquad (9)$$

$$f_{rPF} = 1.69v_f^2 + 2.8v_f + 9.37 \ for \ polypropylene \ fibers \ with \ heat-cool \ cycles \qquad (10)$$

$$f_{rPF} = 1.19v_f + 8.23 \ for \ polypropylene \ fibers \ without \ heat-cool \ cycles \qquad (11)$$

$$f_{rSF} = 2.44v_f + 9.11 \ for \ steel \ fibers \ with \ heat-cool \ cycles \qquad (12)$$

$$f_{rSF} = 6.40v_f + 7.28 \ for \ steel \ fibers \ without \ heat-cool \ cycles \qquad (13)$$

where $f_{rDF}$, $f_{rPF}$, and $f_{rSF}$ are the flexural strength (MPa) of date palm, polypropylene, and steel fiber-reinforced concrete specimens, respectively.

*4.3. Density*

Density is an essential characteristic for high-strength fiber-reinforced concrete. "The density of fiber-reinforced concrete relying on the ingredients used to manufacture it [56]. The density of high-strength concrete containing of date palm, polypropylene, and steel fibers under application of the heat-cool cycles (with and without) as presented in Figure 7a. The density of fibrous high-strength concrete progressively reduces with the increasing the date palm and polypropylene fibers without applying the heat-cool cycles, whereas the density predominantly improves for steel fibers due to the heavier unit of steel fibers over the date palm and polypropylene fibers. In contrast, the densities were gradually reduced by adding different amounts of fiber for date palm, polypropylene, and steel fibers except for the SF1.0 specimen in the application of the heat-cool cycles on the specimens". The density increased up to 3% and 4% by adding steel fibers in the high-strength concrete incorporating heat–cool cycles and without heat–cool cycles, respectively.

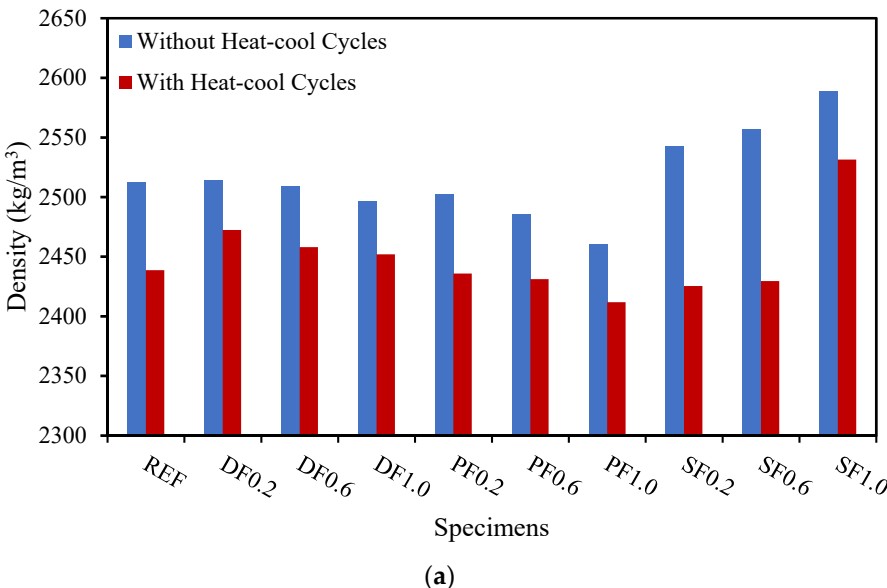

(**a**)

**Figure 7.** *Cont.*

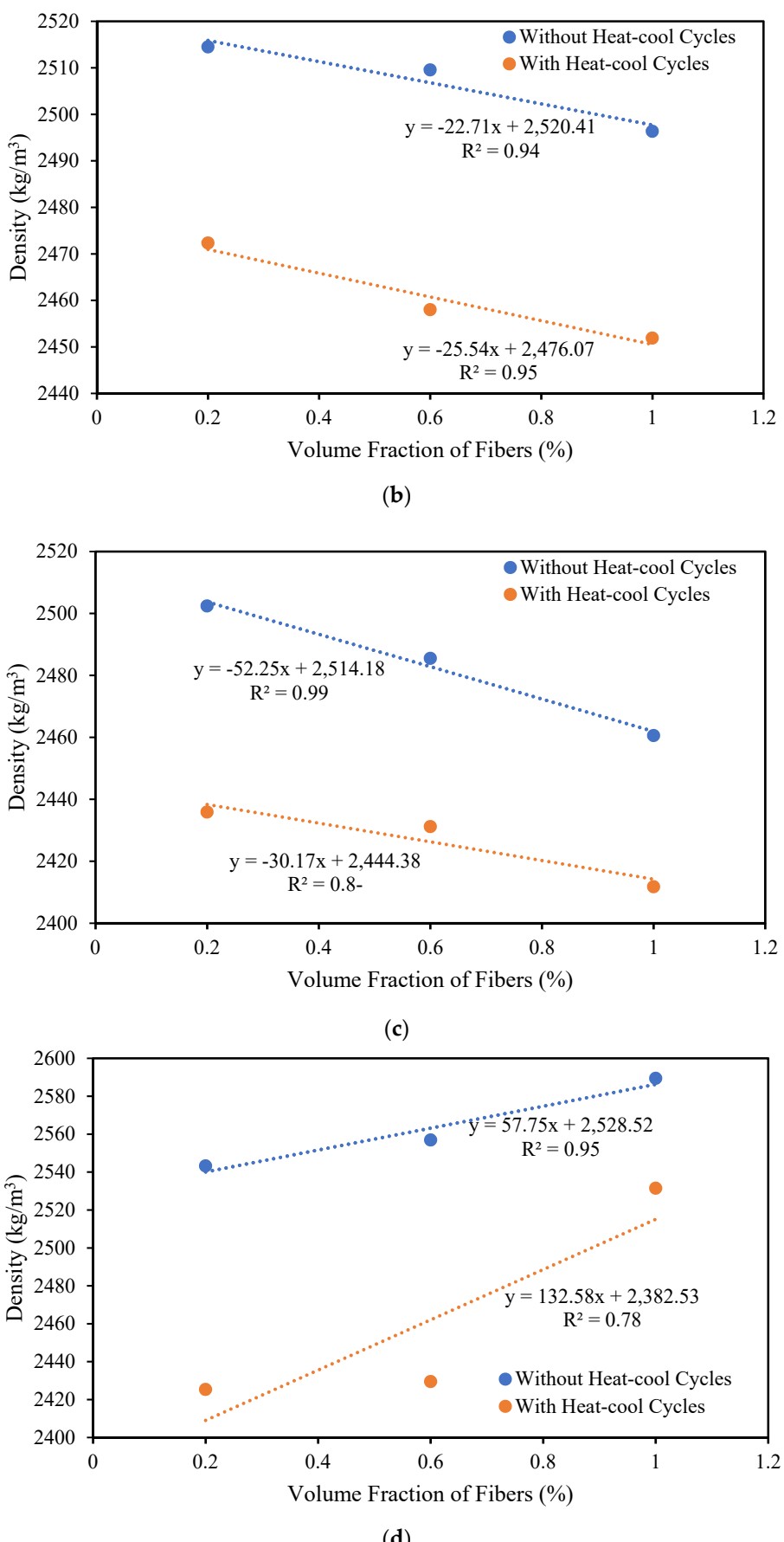

**Figure 7.** Effect of heat–cooling cycles on the density of high-strength fiber-reinforced concrete.

(**a**) Density of fiber reinforced concrete reflecting due to heat–cool cycles; (**b**) Correlation between the density and amount of date palm fibers; (**c**) Correlation between the density and amount of polypropylene fibers; (**d**) Correlation between the density and amount of steel fibers.

A linear correlation emerged between the density and amount of date palm, polypropylene, and steel fiber content for the employed of heat–cool cycles and without heat–cool cycles with strong R$^2$ values, as shown in Figure 7b–d. The density of high-strength concrete comprising date palm, polypropylene, and steel fibers with and without heat–cool cycles might be predicted by the following equations.

$$\gamma_{DF} = -25.54 v_f + 2476.07 \ for \ date \ palm \ fibers \ with \ heat - cool \ cycles \tag{14}$$

$$\gamma_{DF} = -22.71 v_f + 2520.41 \ for \ date \ palm \ fibers \ without \ heat - cool \ cycles \tag{15}$$

$$\gamma_{PF} = -30.17 v_f + 2444.38 \ for \ polypropylene \ fibers \ with \ heat - cool \ cycles \tag{16}$$

$$\gamma_{PF} = -52.25 v_f + 2514.18 \ for \ polypropylene \ fibers \ without \ heat - cool \ cycles \tag{17}$$

$$\gamma_{SF} = 132.58 v_f + 2382.53 \ for \ steel \ fibers \ with \ heat - cool \ cycles \tag{18}$$

$$\gamma_{SF} = 57.75 v_f + 2528.52 \ for \ steel \ fibers \ without \ heat - cool \ cycles \tag{19}$$

where $\gamma_{DF}$, $\gamma_{PF}$, and $\gamma_{SF}$ are the density (kg/m$^3$) of date palm, polypropylene, and steel fiber-reinforced concrete specimens, respectively.

### 4.4. Water Absorption Capacity

The concrete pore structure is known to play a significant role in the material's durability. The amount of water absorbed by immersion provides an estimate of the concrete's total pore volume [57]. The water absorption capacity of the high-strength date palm, polypropylene, and steel fiber-reinforced concrete was higher than the reference concrete specimens in applied heat–cool cycles and without heat–cool cycles. The water absorption capacity was intensified with the increasing amount of fiber in the concrete in both cases, other than the SF0.2 specimen, as shown in Figure 8a. Since the fibers flow around the mortar and the fibers created a connection with them, thus, those fibers increased the micro-pour inside the concrete. As a consequence, the water absorption capacity of fiber-reinforced was increased significantly compared with the reference specimen. However, the date palm, polypropylene, and steel fiber-reinforced concrete exhibited much lower water absorption capacity since the water absorption capacity in good-grade concrete should have to be lower than 10% by weight [58].

The correlation between the water absorption capacity and amount of date palm, polypropylene, and steel fiber content, respectively for the application of heat–cool cycles and without heat–cool cycles with strong R$^2$ values as revealed in Figure 8b–d. The water absorption capacity of high-strength concrete encompassing date palm, polypropylene, and steel fibers with and without heat–cool cycles might be projected by the subsequent equations.

$$W_{cDF} = 0.24 v_f + 1.70 \ for \ date \ palm \ fibers \ with \ heat - cool \ cycles \tag{20}$$

$$W_{cDF} = 0.14 v_f + 1.64 \ for \ date \ palm \ fibers \ without \ heat - cool \ cycles \tag{21}$$

$$W_{cPF} = 0.29 v_f + 1.83 \ for \ polypropylene \ fibers \ with \ heat - cool \ cycles \tag{22}$$

$$W_{cPF} = 0.15 v_f + 1.90 \ for \ polypropylene \ fibers \ without \ heat - cool \ cycles \tag{23}$$

$$W_{cSF} = 0.25 v_f + 1.55 \ for \ steel \ fibers \ with \ heat - cool \ cycles \tag{24}$$

$$W_{cSF} = 0.25 v_f + 1.51 \ for \ steel \ fibers \ without \ heat - cool \ cycles \tag{25}$$

where $W_{cDF}$, $W_{cPF}$, and $W_{cSF}$ are the water absorption capacity (%) of date palm, polypropylene, and steel fiber-reinforced concrete specimens, respectively.

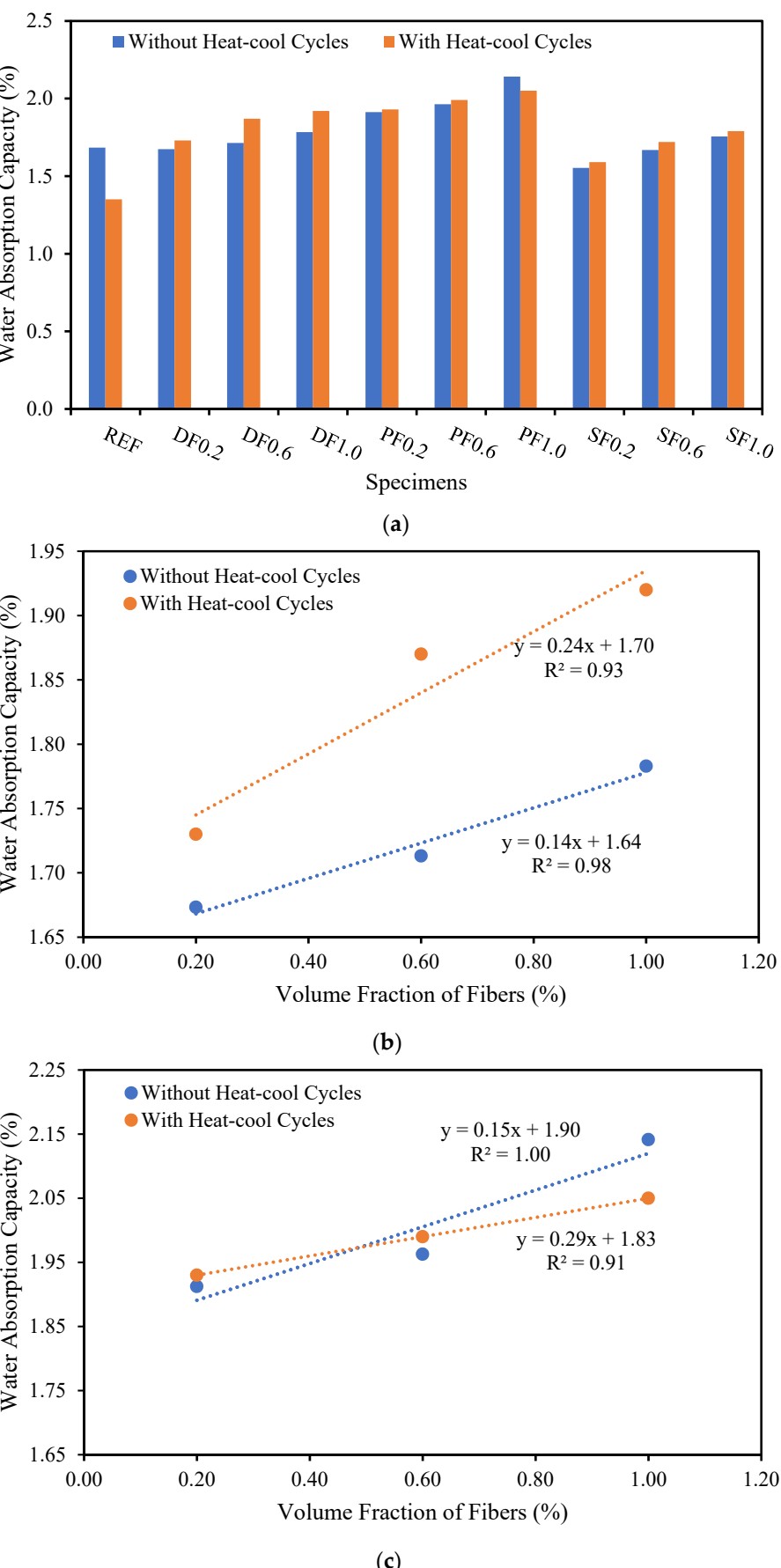

**Figure 8.** *Cont*.

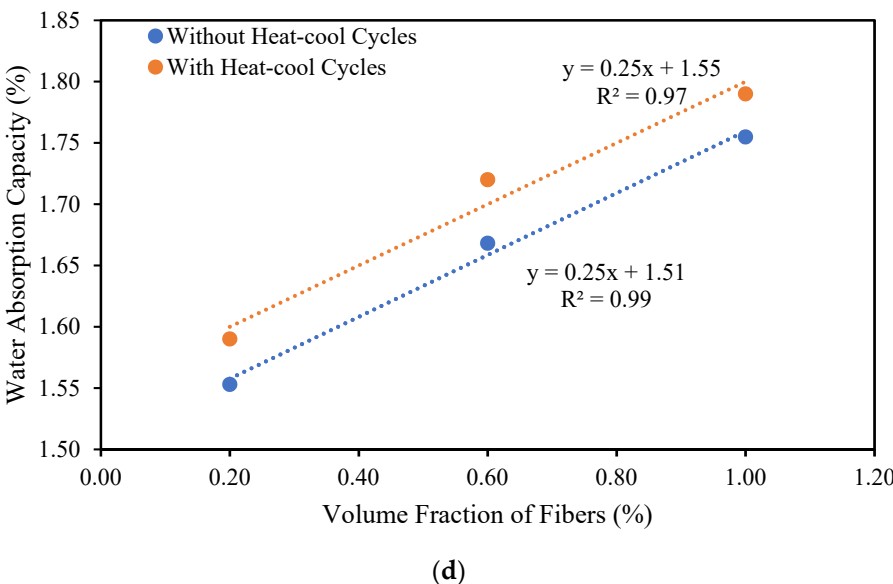

**(d)**

**Figure 8.** Variation of water absorption capacity of fibrous concrete due to heat–cool cycles. (**a**) Variation of water absorption capacity due to heat–cool cycles; (**b**) Correlation between the water absorption capacity and amount of date palm fibers; (**c**) Correlation between the water absorption capacity and amount of polypropylene fibers; (**d**) Correlation between the water absorption capacity and amount of steel fibers.

*4.5. Ultrasonic Pulse Velocity*

One of the most well-known non-destructive methods for evaluating the properties of concrete is UPV [59,60]. "The travel time between the first set and the acceptance of the pulse when the ultrasonic pulse went through the concrete specimen is used to construct the UPV method. The travel route distance between transducers can be used to determine the average wave propagation velocity [61]. Figure 9a depicts the UPV of the high-strength concrete comprising date palm, polypropylene, and steel fibers with and without heat-cool cycles. The date palm, polypropylene, and steel fibers were added to the high-strength concrete with increasing amounts, which helped to reduce ultrasonic wave travel farther and through greater efficiency without heat-cool cycles. There was no influence of heat-cool cycles on the date palm, polypropylene, and steel fibers added to the high-strength concrete with increasing fiber quantities". This was done to ensure the uniformity and homogeneity of this high-strength fiber-reinforced concrete.

The correlation between the UPV and volume fraction of date palm, polypropylene, and steel fiber for the application of heat–cool cycles and without heat–cool cycles as shown in Figure 9b–d. The UPV of high-strength concrete incorporating date palm, polypropylene, and steel fibers with and without heat–cool cycles would be estimated by successive equations.

$$U_{vDF} = 1562.50v_f^2 - 22.50v_f + 4.71 \; for \; date \; palm \; fibers \; with \; heat - cool \; cycles \quad (26)$$

$$U_{vDF} = -13.50v_f + 5.07 \; for \; date \; palm \; fibers \; without \; heat - cool \; cycles \quad (27)$$

$$U_{vPF} = -15.00v_f + 4.67 \; for \; polypropylene \; fibers \; with \; heat - cool \; cycles \quad (28)$$

$$U_{vPF} = -11.49v_f + 4.86 \; for \; polypropylene \; fibers \; without \; heat - cool \; cycles \quad (29)$$

$$U_{vSF} = -15937.50v_f^2 + 130v_f + 4.50 \; for \; steel \; fibers \; with \; heat - cool \; cycles \quad (30)$$

$$U_{vSF} = -35.74v_f + 5.16 \; for \; steel \; fibers \; without \; heat - cool \; cycles \quad (31)$$

where $U_{vDF}$, $U_{vPF}$, and $U_{vSF}$ are the UPV (km/s) of date palm, polypropylene, and steel fiber-reinforced high-strength concrete specimens, respectively.

The mathematical equations are very useful for the different mix proportions of fiber-reinforced high-strength concrete, especially when many variables are used. The outcomes of the fiber-reinforced, high-strength concrete might be obtained without the need to conduct experimental investigations or other field studies. It also might be useful to investigate the effects of one or more variables on the performance of high-strength concrete.

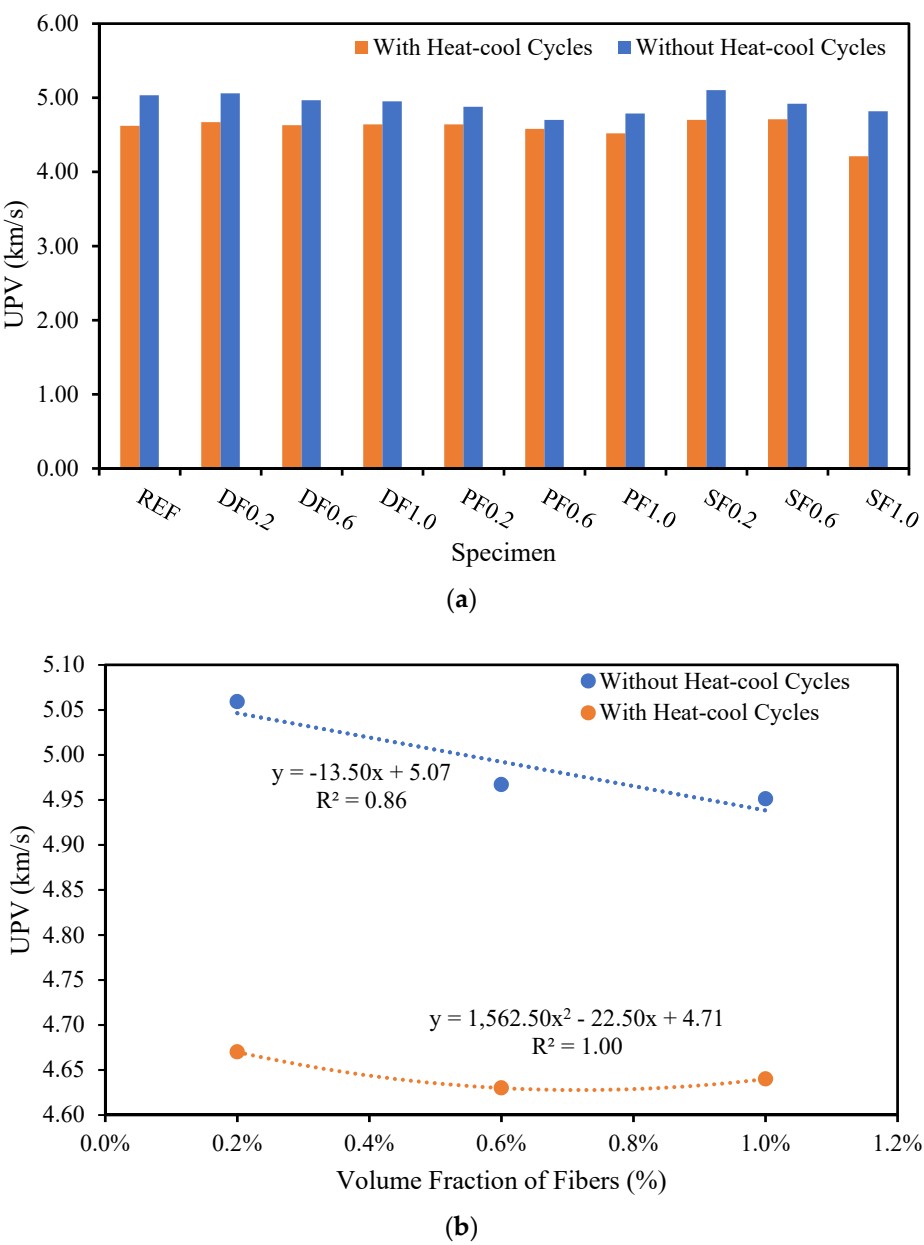

**Figure 9.** *Cont.*

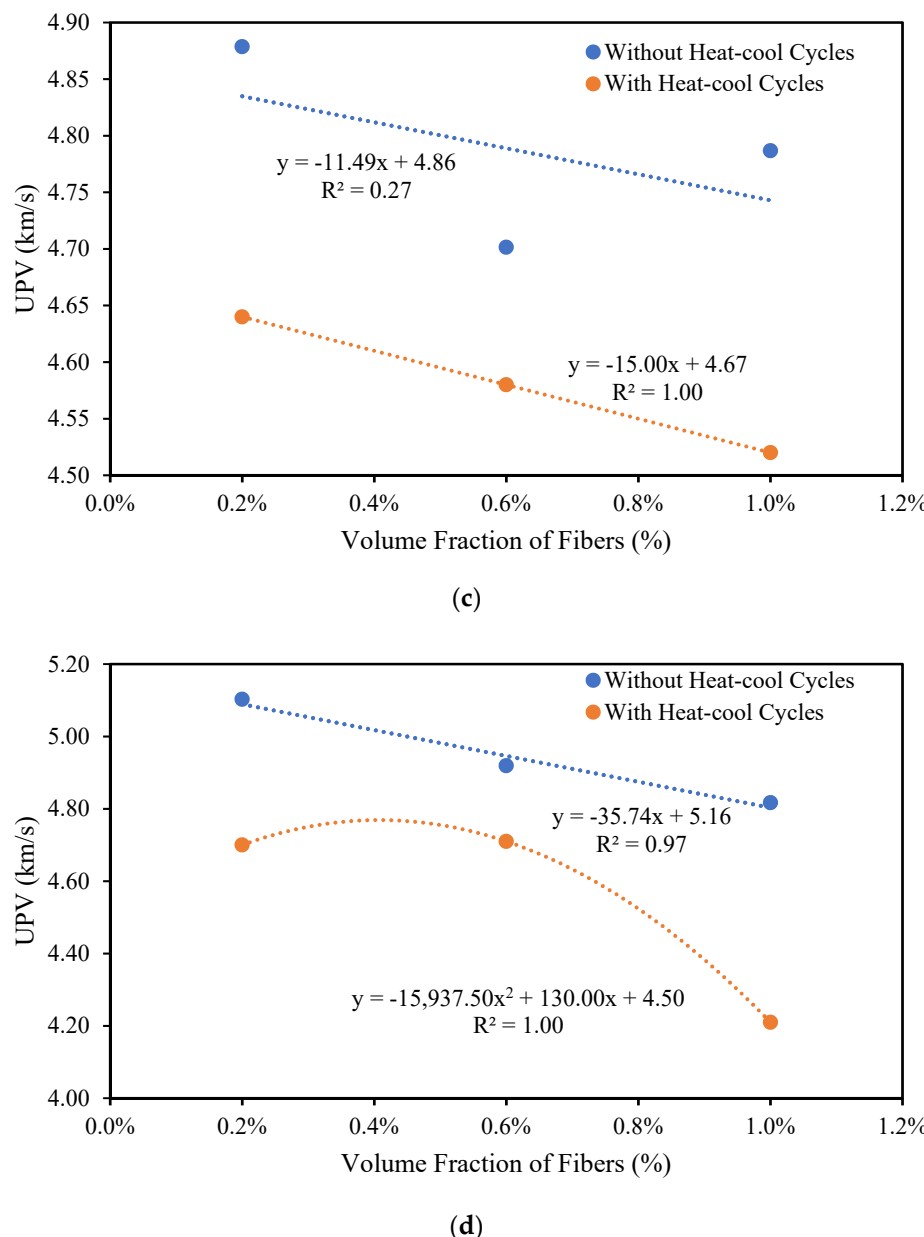

**Figure 9.** Deviations of UPV due to heat–cooling for high-strength fiber-reinforced concrete. (**a**) UPV of the fibrous concrete changed due to heat–cool cycles; (**b**) Correlation between the UPV and amount of date palm fibers; (**c**) Correlation between the UPV and amount of polypropylene fibers; (**d**) Correlation between the UPV and amount of steel fibers.

*4.6. Energy Absorption Capacity*

The energy absorption capacities and their enhancement for high-strength date palm, polypropylene, and steel fiber-reinforced concrete specimens without heat–cool cycles are demonstrated in Figure 10a. The use of date palm, polypropylene, and steel fibers revealed an enhancement in the energy absorption capacity of up to 2%, 2.5%, and 39%, respectively, compared with the reference specimen when no heat–cool cycles were applied. Increasing the amount of fiber reinforcement in the concrete mixes progressively enhanced the load-bearing capacity and energy absorption capacities.

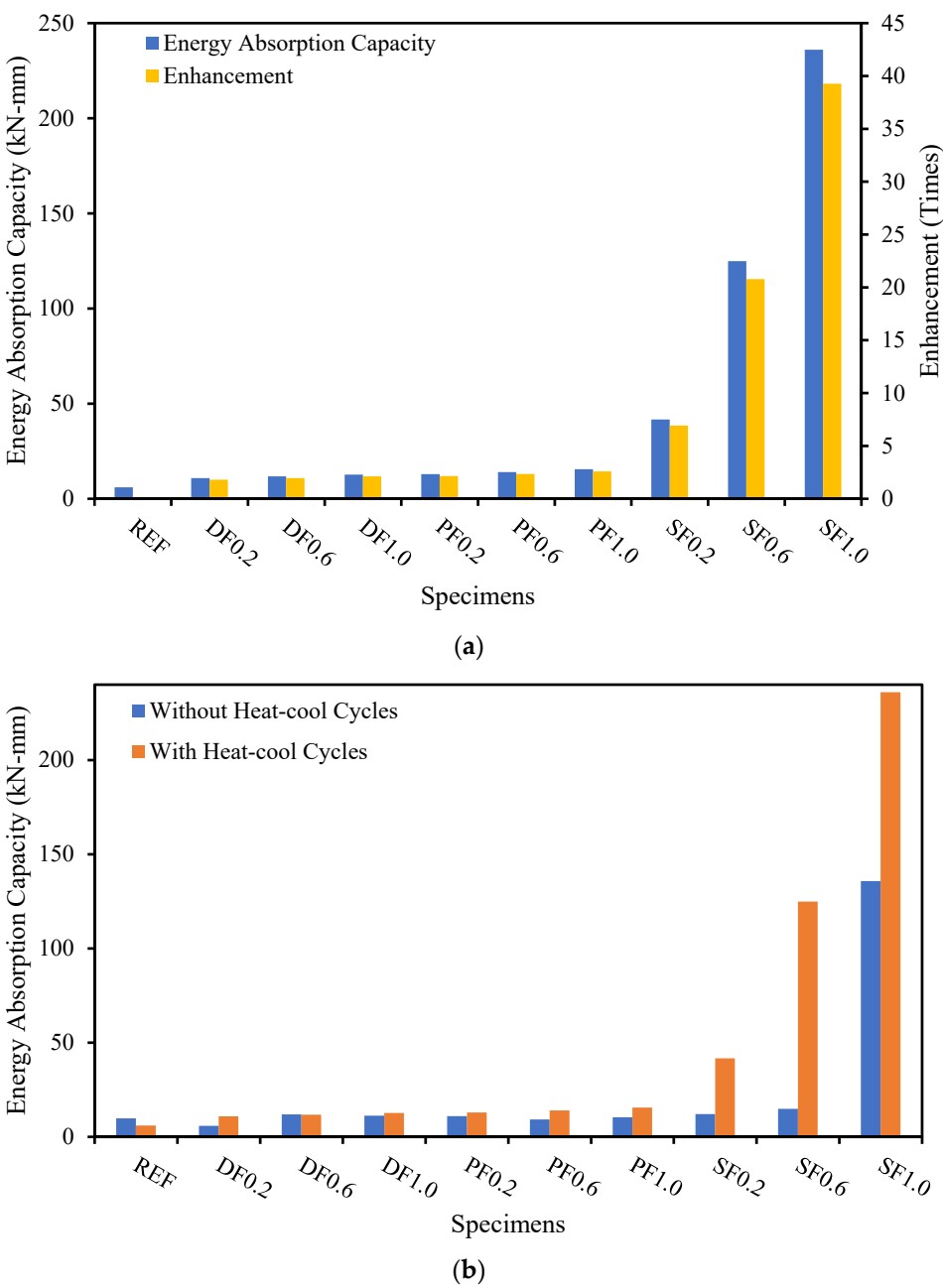

**Figure 10.** Performance energy absorption capacity of high-strength concrete due to heat–cooling cycles. (**a**) Energy absorption capacity and its enhancement without heat–cool cycles; (**b**) Energy absorption capacity comparison without and with heat–cool cycles.

The high-strength date palm, polypropylene, and steel fiber-reinforced concrete specimen's energy absorption capacities with and without heat–cool cycles are shown in Figure 10b. According to the assessment of the ruptured failure specimens, the crack-bridging influence between the fibers and the grout matrix triggered the failure. As a consequence, the steel fibers exhibited greater energy absorption in both cases (with and without heat–cool cycles) over the date palm and polypropylene fiber-reinforced concrete specimens since the steel fiber had a higher stiffness. Therefore, it should be suggested that the concrete with steel fibers is more applicable in harsh weathering action compared with the date palm and polypropylene fiber concrete specimens.

## 5. Conclusions

The influence of heat–cool cycles on high-strength concrete reinforced with various percentages (0%, 0.2%, 0.6%, and 1.0%) of date palm, polypropylene, and steel fibers during manufacturing was investigated. The engineering properties were examined, including compressive strength, flexural strength, density, water absorption capacity, ultrasonic pulse velocity, and energy absorption capacity. The following concluding remarks can be illustrated based on the laboratory examination:

The compressive strength of the high-strength concrete comprising date palm, polypropylene, and steel fibers was enhanced significantly with increasing fiber contents without implementing the heat–cool cycles. By contrast, compressive strength was reduced by applying the heat–cool cycles on the specimens containing date palm and polypropylene fibers; however, the steel fibers encompassing specimens expressed a substantial improvement in this case, which was due to the higher compressive load-bearing capacity of the fibers.

The flexural strength substantially improved with increasing the date palm, polypropylene, and steel fibers into the high-strength concrete with and without heat–cool cycles. Increasing the date palm, polypropylene, and steel fibers from 0% to 1.0% enhanced the flexural strength up to 85%, 79%, and 165%, respectively, compared with the reference specimen without the implementation of heat–cool cycles. On the other hand, the flexural strength improved up to 4%, 2%, and 34%, respectively, over the reference specimens with the implementation of heat–cool cycles.

The density was gradually reduced as the date palm and polypropylene fibers increased without applying the heat–cool cycles, whereas the density for steel fibers was noticeably improved by the steel fibers' heavier unit compared to the date palm and polypropylene fibers. In contrast, the densities were decreased by adding the different amounts of fiber for date palm, polypropylene, and steel fibers, except for the SF1.0 specimen in the application of the heat–cool cycles.

The water absorption capacity was increased with the increasing amounts of date palm, polypropylene, and steel fibers into high-strength concrete as fibers increased the micropore in the concrete both with and without the implementation of heat–cool cycles.

With the addition of the date palm, polypropylene, and steel fibers, the high-strength concrete exhibited an improvement in UPV compared with the reference specimen without heat–cool cycles applied. A negligible impact was observed with the implementation of heat–cool cycles on the date palm, polypropylene, and steel fiber-reinforced high-strength concrete specimens.

The addition of date palm, polypropylene, and steel fibers into high-strength concrete showed a substantial improvement in energy absorption capacity compared with the reference specimen in the cases both with and without the implementation of heat–cool cycles.

Therefore, the natural date palm fibers might be used to produce sustainable fibrous high-strength concrete and be applicable in severe weathering conditions.

This study only applied 60 °C for thermal cycles on the specimens and afterward, specimens were cooled for two days at room temperature $25 \pm 5$ °C (1 cycle) for 180 days.

**Author Contributions:** Conceptualization, I.H., M.A.H., M.A., S.Q. and Y.Ö.; methodology, I.H., M.A.H., M.A., S.Q. and Y.Ö.; software, I.H., M.A.H., M.A., S.Q. and Y.Ö.; validation, I.H., M.A.H., M.A., S.Q. and Y.Ö.; formal analysis, I.H., M.A.H., M.A., S.Q. and Y.Ö.; investigation, I.H., M.A.H., M.A., S.Q. and Y.Ö.; resources, I.H., M.A.H., M.A., S.Q. and Y.Ö.; data curation, I.H., M.A.H., M.A., S.Q. and Y.Ö.; writing—original draft preparation, I.H., M.A.H., M.A., S.Q. and Y.Ö.; writing—review and editing, I.H., M.A.H., M.A., S.Q. and Y.Ö.; visualization, I.H., M.A.H., M.A., S.Q. and Y.Ö.; supervision, I.H., M.A.H., M.A., S.Q. and Y.Ö.; project administration, I.H., M.A.H., M.A., S.Q. and Y.Ö.; funding acquisition, I.H., M.A.H., M.A., S.Q. and Y.Ö. All authors have read and agreed to the published version of the manuscript.

**Funding:** This research was funded by the Deanship of Scientific Research at Najran University with grant number (NU/NRP/SERC/11/26).

**Institutional Review Board Statement:** Not applicable.

**Informed Consent Statement:** Not applicable.

**Data Availability Statement:** Not applicable.

**Acknowledgments:** The authors are thankful to the Deanship of Scientific Research at Najran University for funding this work under the National Research Priorities funding program grant code (NU/NRP/SERC/11/26).

**Conflicts of Interest:** The authors declare no conflict of interest.

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
