# Peer review of "Influence of Heat–Cool Cyclic Exposure on the Performance of Fiber-Reinforced High-Strength Concrete"

_sustainability, doi:10.3390/su15021433_

Round 1

Reviewer 1 Report

This paper presents an experimental statistical study of the influence of heat-cool cycles on high-strength concrete comprising various fibers (natural date palm, polypropylene, and steel fibers) and their different volume percentages. Compressive strength, flexural strength, density, water absorption capacity, ultrasonic pulse velocity, and energy absorption capacity were analyzed.

The paper topic is interesting and the author has done a lot of work. In this regard, I am supportive of its acceptance. Thus, a major revision is requested.  

There are some details that I would like the authors to provide:

1.  The state of the art is extensive, the authors analyzed 59 works. However, the paper below is not analyzed.

Althoey, F.; Hakeem, I.Y.; Hosen, M.A.; Qaidi, S.; Isleem, H.F.; Hadidi, H.; Shahapurkar, K.; Ahmad,J.; Ali, E. Behavior of Concrete Reinforced with Date Palm Fibers. Materials 2022, 15, 7923.https://doi.org/10.3390/ma15227923

From my point of view, the authors should not present the research already published but only refer to the work already published.

For example, Figure 3 shows Polypropylene fibers. These figure have been presented in the above paper.

The present work must be delimited from the previous work. The results obtained after heat-cool cycles are compared with those without heat-cool cycles. Are the results obtained without heat-cool cycles previously published?

I noticed some differences between the results published in the previous article and those analyzed in this article, although the analyzed samples have the same structure.

 I don't understand. Please clarify this aspect and clearly delimit the two publications.

The research presented and analyzed in the paper already published should be delimited by those to be published in Sustainability.

2.  The title is too long, the title must reflect the content of the work in a few words

3.  It is not clear whether the properties of date palm fibers, polypropylene fibers, and steel fibers presented in tables 3, 4, and 5 were determined by the authors or are taken from the bibliography.

4.  How were the fiber percentages chosen? Why weren't 0.5%, 1%, and 1.5% chosen?

5.  It is well known that in order to have confidence in experimental results, repetitions are necessary. Have the experiments been repeated and a statistical analysis developed? If not, the experiments should be repeated and average values should be used. Equations determined using a single experiment cannot be trusted. At the same time, a statistical analysis is required.

6.  The predicted flexural strength equations for steel fiber reinforced concrete specimens with implementing heat-cooling cycles have correlations of 0.5. It is probably necessary to look for another form of the equation (it is not linear). The same observation for the correlation between UPV and the amount of polypropylene fibers, for the correlation between UPV and amount of date palm fibers.

7.  The presented correlations are not analyzed. A much stronger discussion and analysis are warranted. Lacking appropriate analysis and discussion of the results presented, it appears that there is relatively little useful information for other researchers who might be interested in this area.

While I am unable to recommend publication at this time, I believe that the authors should revise the manuscript taking the comments into account.

Author Response

Dear Reviewer,

Thanks for all the time and effort you put into reviewing our paper. This is a great contribution to the scientific community. It's much appreciated. Thanks.

Regarding the comments, you will find the authors' responses in the attachments. Also highlighted in the revised manuscript.

Kind Regards,

Reviewer 2 Report

The authors have submitted a well prepared paper on the interesting topic of the Influence of Heat-cool Cyclic Exposure on the Performance of High-Strength Concrete Comprising Date Palms, polypropylene and Steel Fibers. The highlights in this study, the influence of heat-cool cycles on high-strength concrete comprising various fibers such as natural date palm, polypropylene, and steel fibers and their different volume percentages. The paper is clearly presented and provides interesting results. This study is valuable for the practical engineering. However, the following comments are provided to assist the authors to improve the paper:

1) The article's purpose should be clarified in detail, why this study could be beneficial, and a more in-depth conclusion in applications should be provided.

2) 1. Introduction; I recommend expanding to ref [24] the following articles related to polypropylene fiber. https://doi.org/10.3390/su14116839

3) 2.1.4.1. Chemical Treatment of Date Palm Fibers; "a treatment with 3% NaOH was chosen due to the highest tensile strength of the fibers" Why does the 3 molar concentration give the highest tensile strength of the fiber? Please explain.

4) Table 3; Why is the diameter of date palm fiber at 6.0% concentration NaOH greater than that at 1.0% and 3.0% concentration NaOH? Please describe.

5) Why the fiber contents (date palm, polypropylene, and steel) blended with concrete with varying amounts of 0.2%, 0.6%, and 1.0% by volume of concrete were used? Please explain.

6) Figure 5(a); Improved the position of the dotted lines.

7) Equation 2 to Equation 31 should indicate the scope of the application.

8) The sample considered for this study needs to be more convincing. What is the standard deviation and COV value considered?

9) The results of high-strength concrete using diverse fibers are highly dependent on the inputs provided/assumed or resources of raw materials, hence the obtained results in the study should be compared with the existing literature.

10) Conclusions: the author should further explain this research's construction application limitations. Please describe in the conclusion.

Author Response

(The authors gave the same response as above.)

Round 2

Reviewer 1 Report

Dear authors,

Thanks for your answers.

At this time, I consider that you have completed the manuscript taking into account most of my observations.

Regarding the answers to observations 1,5 and 6, I didn't quite understand your entire justification.

Observation 1:

" The state of the art is extensive, the authors analyzed 59 works. However, the paper below is not analyzed.

Althoey, F.; Hakeem, I.Y.; Hosen, M.A.; Qaidi, S.; Isleem, H.F.; Hadidi, H.; Shahapurkar, K.; Ahmad,J.; Ali, E. Behavior of Concrete Reinforced with Date Palm Fibers. Materials 2022, 15, 7923.https://doi.org/10.3390/ma15227923

From my point of view, the authors should not present the research already published, but only refer to the work already published.

For example, Figure 3 shows Polypropylene fibers. These figure have been presented in the above paper.

The present work must be delimited from the previous work. The results obtained after heat-cool cycles are compared with those without heat-cool cycles. Are the results obtained without heat-cool cycles previously published?

I noticed some differences between the results published in the previous article and those analyzed in this article, although the analyzed samples have the same structure.

 I don't understand. Please clarify this aspect and clearly delimit the two publications.

The research presented and analyzed in the paper already published should be delimited by those to be published in Sustainability."

The answer to observation 1:

“Yes, we have published the paper which contains only 28 days of results. This article contains totally new study of the effect of heat-cool cycles on the specimens over the period of 180 days and 28 days results are used only for reference. Therefore, the outcomes of this study are totally different and analyzed also differently presented.”

I would have appreciated a response related to your research.

Although the authors claim that "the results are used for reference only", the paper from which the data are taken does not appear in the bibliography.

From my point of view, the work must be analyzed in the bibliographic study (in the introduction).

I understand that the research in this paper is compared to that in the already-published paper (as a reference). However, the published work does not appear in the bibliography, it is not analyzed in the introduction.

I ask the authors to refer to the results from the already published work and to check the data used.

Observation 5:

“It is well known that in order to have confidence in experimental results, repetitions are necessary. Have the experiments been repeated and a statistical analysis developed? If not, the experiments should be repeated and average values should be used. Equations determined using a single experiment cannot be trusted. At the same time, a statistical analysis is required.

The answer to observation 5:

Each mixture of the concrete was contained three specimens and average values are presented in this study. This project is ongoing and hence the statistical analysis did not add to this manuscript.”  

I did not find where the authors specified that the presented results are average and that three samples were analyzed in each case. In the design of the experiments, it is not specified whether all were repeated or only certain experiments and how many times they were repeated.

I ask the authors to specify these aspects in the paper.

Observation 6:

The predicted flexural strength equations for steel fiber reinforced concrete specimens with implementing heat-cooling cycles have correlations of 0.5. It is probably necessary to look for another form of the equation (it is not linear). The same observation for the correlation between UPV and the amount of polypropylene fibers, for the correlation between UPV and amount of date palm fibers.

The answer to observation 6:

The predicted equations have been modified based on the comments in the revised manuscript.”

The proposed equations must be checked if they are adequate to represent the experimental data (for example with the F-test).

For example, if the correlation is 0.78 or 0.27, is the model correctly chosen? please indicate a bibliographic source that recommends this.

In conclusion, I appreciate the work done and recommend the publication of the paper in Sustainability after revisions.

Author Response

(The authors gave the same response as above.)

Reviewer 2 Report

Dear authors, thank you for the revised version. It has been well-revised based on the reviewer's comments and can be accepted for publication.

Author Response

Dear Reviewer,

Thanks for all the time and effort you put into reviewing our paper. This is a great contribution to the scientific community. It's much appreciated. Thanks.

Kind Regards,